# Evolutionary Trajectories of Primary and Metastatic Pancreatic Neuroendocrine Tumors Based on Genomic Variations

**DOI:** 10.3390/genes13091588

**Published:** 2022-09-04

**Authors:** Midie Xu, Jiuliang Yan, Beiyuan Hu, Chuntao Wu, Haitao Gu, Zihao Qi, Tao Chen, Wenting Yang, Yan Zheng, Hanguang Dong, Weiqi Sheng, Jiang Long

**Affiliations:** 1Department of Pathology, Fudan University Shanghai Cancer Center, Shanghai 200032, China; 2Department of Medical Oncology, Shanghai Medical College, Fudan University, Shanghai 200032, China; 3Institute of Pathology, Fudan University, Shanghai 200032, China; 4Department of Pancreatic Surgery, Shanghai General Hospital, Shanghai Jiao Tong University School of Medicine, Shanghai 200080, China; 5Shanghai Key Laboratory of Pancreatic Disease, Institute of Pancreatic Disease, Shanghai Jiao Tong University School of Medicine, Shanghai 200080, China; 6Department of Genetics, Stanford University, Stanford, CA 94305, USA

**Keywords:** pancreatic neuroendocrine tumors, liver metastasis, genomic heterogeneity, tyrosine kinase, DNA repair

## Abstract

Liver metastases are common in pancreatic neuroendocrine tumors (PanNETs) patients and they are considered a poor prognostic marker. This study aims to analyze the spatiotemporal patterns of genomic variations between primary and metastatic tumors, and to identify the key related biomolecular pathways. We performed next-generation sequencing on paired tissue specimens of primary PanNETs (*n* = 11) and liver metastases (*n* = 12). Low genomic heterogeneity between primary PanNETs and liver metastases was observed. Genomic analysis provided evidence that polyclonal seeding is a prevalent event during metastatic progression, and may be associated with the progression-free survival. Besides this, copy number variations of BRCA1/BRCA2 seem to be associated with better prognosis. Pathways analysis showed that pathways in cancer, DNA repair, and cell cycle regulation-related pathways were significantly enriched in primary PanNETs and liver metastases. The study has shown a high concordance of gene mutations between the primary tumor and its metastases and the shared gene mutations may occur during oncogenesis and predates liver metastasis, suggesting an earlier onset of metastasis in patients with PanNETs, providing novel insight into genetic changes in metastatic tumors of PanNETs.

## 1. Introduction

Pancreatic neuroendocrine tumors (PanNETs) are extremely rare and account for less than 2% of all pancreatic tumors, with an estimated annual incidence of less than 1/100,000 per year [1,2]. The cell of origin of this disease remains debated, and several recent studies suggest that they are most probably derived from islet cells of the pancreas [3]. According to the World Health Organization (WHO) 2010 Grading System, PanNETs are categorized into Grade 1 (mitotic counts per 10 HPF < 2, Ki-67 index ≤ 2%), Grade 2 (mitotic counts per 10 HPF 2-20, Ki-67 index 3–20%), and Grade 3 (mitotic counts per 10 HPF > 20, Ki-67 index > 20%). Collectively, PanNETs constitute a heterogeneous group of tumors with a wide spectrum of clinicopathological features that have unpredictable clinical manifestations, which often obfuscate unambiguous diagnosis [4]. In order to gain a better understanding of the molecular ontogeny of this heterogeneous disease, Scarpa et al. utilized whole-genome sequencing to interrogate the somatic mutation landscape in 102 primary PanNETs, and unraveled a large number of genetic alterations associated with aberrant phosphoinositide 3-kinase (PI3K) and mammalian target of rapamycin (mTOR) signalings, dysregulated cell cycle/proliferation, DNA damage response, chromatin remodeling/histone methylation, and telomere alterations [5]. The mutation profiles may potentially be useful in defining clinically relevant subtypes for the purpose of risk stratification. To date, physicians still face challenges in identifying the aggressive subtypes due to the lack of understanding in the molecular mechanisms underlying disease progression from primary tumors to distant metastasis. For example, 20% of patients subtyped by high Ki67 proliferation rate progressed to metastatic disease after treatment [6]. More than 50% of PanNETs patients are diagnosed at advanced stages with lymph node, liver, or distant metastasis. The 5-year survival rate for patients with untreated liver metastases ranges from 20% to 40% [7]. Understanding the genetic basis of disease progression would thus allow physicians to cater personalized treatments to achieve a more positive outcome.

The genetic diversity in metastatic lesions is generally much less explored compared to primary tumors. Given that metastasis is a late event during malignant disease progression, it is typically thought to be seeded by a small founder population from the primary tumor [8]. However, several models of metastatic progression have been proposed, each comes with different implications for clinical management of the disease. The prevailing linear acquisition model posits that primary tumors gradually gain metastatic potential as they acquire more somatic mutations at later stage of the disease [9], while the alternative model suggests that some tumors are ‘born-to-be-bad’ with metastatic potential conferred by specific mutations present early in the primary tumor cells [10]. In addition, metastatic lesions could be formed by polyclonal seeding from several subclonal populations of primary tumor cells, or by monoclonal seeding with a single genetic population. Thus, the mode of metastatic disease progression dictates the resulting genetic heterogeneity and the underlying driver alterations in the metastatic lesions. Hitherto, the molecular events driving distant metastasis of PanNETs remain largely unknown. Particularly, it is unclear whether the fate of metastasis is predestined at the time of tumor initiation in PanNETs, and whether the metastatic lesions are seeded by a single or multiple genetic population(s) from the primary site.

In the current study, we leveraged matched primary-metastatic tumors to interrogate the genetics of metastasis in PanNETs. Herein, we performed targeted next-generation sequencing (NGS) using a 468-gene panel on the matched tumor pairs from patients who underwent debulk surgery. We aimed to analyze the spatiotemporal patterns of genomic variations and to identify the key molecular pathways in the metastatic process. In addition, we performed whole-exome sequencing (WES) on the primary tumor, liver metastasis, and ovarian metastasis of one PanNET patient to further elucidate the evolutionary trajectory.

## 2. Materials and Methods

This study was approved by the institutional review board of Fudan University Shanghai Cancer Center (FUSCC). Tissue acquisition was carried out in accordance with Institutional and State guidelines on the experimental use of human tissues.

### 2.1. Tissue Acquisition

Paired snap frozen or paraffin-embedded tissue specimens of primary PanNETs (*n* = 11) and liver metastases (*n* = 12) were acquired from eleven treatment-naïve patients with pathologically proven sporadic PanNETs who underwent debulk surgery at FUSCC from February 2014 to July 2019. In addition, ovarian metastases (*n* = 2) were obtained from one patient apart from acquisition of tissue specimens of primary PanNET and liver metastases. 

### 2.2. DNA Extraction and Sequencing

Six pairs of fresh tissue specimens of primary tumors and liver metastases were paraffin-embedded according to conventional protocols and stored at room temperature, and five additional pairs of primary tumors and liver metastases were snap frozen and stored at −80 °C. DNA was extracted from paraffin-embedded tissues using Biospin FFPF Tissue Genomic DNA Extraction Kit (BSC24S1, BioFlux, Beijing, China) according to the manufacturer’s instructions. We extracted DNA from snap-frozen tumors using TIANamp Genomic DNA Kit (DP180123, TianGEN, Beijing, China). All DNA samples were quantified using a NanoDrop (ND 1000, Thermo Scientific, Waltham, MA, USA ). Libraries were generated using NanoPrep™ DNA Library kit (for Illumina^®^) (1002101C1, Nanodigmbio, Nanjing, China) according to the manufacturer’s instructions. Gene sequencing was done using the Illumina NextSeq500 System (Illumina, San Diego, CA, USA). The custom panel test covers a total of 468 cancer-associated genes, including 35 fusion genes (Genenexus Technology Corp., Yangzhou, China). 

### 2.3. Quality Control and Single Nnucleotide Variant (SNV) Analysis

Quality of the raw data was assessed by FastQC software (version 1.0) (Available from: https://qubeshub.org/resources/fastqc, accessed on 28 July 2022). Sequence data were aligned to the human genome assembly GRCH37 using BWA (version 0.7.17) [11] and SAMtools (version 1.16) [12] to generate BAMs. Variant calling was performed using VarDict (version 1.8.2) [13]. Variants were annotated using SNPEff to filter out the common SNPs that are reported by dbSNP or COSMIC database. The predicted function of mutations were identified by SNPeff (version 5.1) [14]. All the analyses were performed under default parameter.

### 2.4. Structural Variant (SV) and Copy Number Analysis

SVs were determined using Delly (version 0.9.1) [15]. SV breakpoints and potential consequence of the SVs were determined by annotation against Ensemble known genes (version 75) using in-house scripts. Copy number was determined using CNVkit (version 0.9.9) [16]. One copy indicated copy number loss (excluding genes on the X chromosome in male patients), zero copies indicated homozygous deletion, and a copy number ≥ 3 indicated copy gain. All the analyses were performed under default parameters.

### 2.5. Gene Set Enrichment Analysis

To identify the main signaling pathways mediating tumorigenesis and liver metastasis of PanNETs, we carried out gene set enrichment analysis based on the categories of gene ontology (GO, Biological Process) and the Kyoto Encyclopedia of Genes and Genomes (KEGG) using the clusterProfiler enriched category, and the significance threshold was set as adjusted *P* value and Q value less than 0.0001. Pathway and network analyses were performed using Ingenuity Pathway Analysis (IPA).

### 2.6. Phylogenetic Tree Reconstruction

To reconstruct the phylogeny of paired primary and metastatic tumors from individual patients based on SNVs and indels, the allele fractions (AF) of SNVs and indels were used as input to calculate tumor subcloning using Clonality Inference in Tumors Using Phylogeny (Citup) (version 0.1.2) [17], and all possible results were inferred and the one with the highest score was selected as the final result.11 Then, the proportion of Citup was used to find the phylogenetic tree and different subclones in the sample were extracted, and the R package TimeScape (version 1.20) was used for visualization to generate fish graph. All the analyses were performed under default parameter.

### 2.7. Statistical Analysis

Progression-free survival (PFS) was calculated from the date of surgery to the first evidence of progressive disease or death from any cause, whichever occurred first. For patients without progressive disease who did not die during the study period, PFS data were censored on the date of the final tumor assessment. PFS was summarized using Kaplan–Meier methods. Hazards ratios (HR) and 95% confidence interval (95% CI) were estimated using a Cox proportional hazards model. Patients were categorized into the high and low genomic heterogeneity group using the median percentage of shared mutations in the primary PanNETs and their liver metastases. Cox proportional hazards model was also used to compare the results obtained in the two groups and in patients with and without BRCA1/2 copy number variations (CNVs). The *p* value of the survival difference was calculated by the log-rank test. All the analyses were conducted in R-3.6.0.

### 2.8. Data Availability 

The data that support the findings of this study are available on request from the corresponding author.

## 3. Results

### 3.1. Patient Demographic and Baseline Characteristics

Patient demographic and baseline characteristics are shown in Table 1. The study included a total of eleven patients who had sporadic PanNET with liver metastasis. Their median age was 53 years (range 33–67 years) and seven patients (63.6%) were female. The pancreatic tumor was located in the head of the pancreas in four patients, in the body in one patient, and in the tail in six patients. Two patients were pathologically classified with grade 1 tumors while nine patients had grade 2 tumors. Ten patients underwent pancreatectomy and hepatectomy simultaneously within one month of diagnosis of PanNETs.

### 3.2. Genomic Landscape of Primary Tumors and Metastases 

Eleven pairs of PanNETs and liver metastases were subjected to gene panel sequencing. The curated data revealed a median of 204 (range 151–405) high-confidence SNVs and insertion-deletions (Indels) in the primary tumors and 197 (range 149–468) in liver metastases (Figure 1A). Interestingly, we observed that eight out of eleven patients had significantly more shared mutations in the tumor pairs than private mutations in either the primary or the metastatic tumors (range: 9.5–74.2%, median: 60.4%, Figure 1A–C). Taken together, these results may suggest that: (1) there is limited heterogeneity of pathogenic driver mutations between paired primary PanNETs and their liver metastases, and (2) the propensity or potential to metastasize may be acquired early during PanNETs tumorigenesis. 

To examine if the degree of shared mutations between primary tumors and their metastases affects the progression-free survival (PFS) of patients, we performed a Kaplan–Meier analysis. Specifically, we defined a patient as having low proportion of shared mutations if <60% of all detected somatic alterations were shared between the primary tumor and the matched metastasis to categorize the patients into two groups (Low group, *n* = 5; High group, *n* = 6). Low degree of shared mutations would indicate that the metastatic lesions had acquired more private mutations that might be necessary to establish niche independence. The Kaplan–Meier analysis showed that the degree of common mutations seemed likely to be associated with the longer PFS(Low group versus high group: 35.0 months, 95% CI, 20.7–49.3 vs. 26.0 months, 95% CI, 20.2–33.5; hazards ratio [HR], 1.346, 95% CI, 0.411–4.411; *p* = 0.188) (Figure 1D). However, there is no significant statistical difference between the two groups, which may be related to the small sample size. Studies with the larger sample size are needed for further validation.

Next, we aimed to identify recurrently mutated genes that were present in the primary-metastatic tumor samples in at least two patients. Fourteen SNVs in thirteen genes were identified, including nine missense mutations, two stop gains, and one frameshift variant: KMT2C (36.4%, 4/11), GNAQ (27.3%, 3/11), LIMK1 (27.3%, 3/11), CSK (18.2%, 2/11), EPHA2 (18.2%, 2/11), FLT3 (18.2%, 2/11), KMT2D (18.2%, 2/11), MEN1 (18.2%, 2/11), NRG3 (18.2%, 2/11), RANBP2 (18.2%, 2/11), ROS1 (18.2%, 2/11), SETD2 (18.2%, 2/11), and TNK2 (18.2%, 2/11) (Figure 2A and Appendix A). Among those genes, KMT2C and MEN1 have been reported to be associated with PanNET [18]. CSK, FLT3, and NRG3, are involved in RAF/MAP kinase cascade and MAPK1/MAPK3 signaling. Besides this, we also observed mutations in other 16 reported PanNET-related genes: APC, ARID2, ATM, BRCA1/2, DAXX, MSH3, MSH6, PALB2, RAD50, RAD51, RB1, SMARCA4, TSC1, and TSC2 (Table 2) [11]. Many of these genes are involved in DNA damage response and repair, including homologous recombination repair and DNA repair, indicating that impairment of multiple DNA damage repair processes in PanNETs and their metastases. 

Then, we performed gene set enrichment analysis to further identify the key molecular mechanisms that have been altered by the genetic alterations. Consistent with previously published studies [19,20,21], we found pathways in cancer were significantly affected in PanNETs and liver metastases, and that EGFR tyrosine kinase inhibitor resistance and transcriptional misregulation in cancer were unique to liver metastases via KEGG pathway analysis; simultaneously, GO Biological Processes analysis underlined those signaling pathways tightly related to tyrosine phosphorylation, DNA repair, and cell cycle regulation, especially in liver metastases (Table 3, Appendix A). 

We also identified seven genes (BRCA1, BRCA2, RANBP2, SPTA1, ATRX, ATM, and LRP1B) with copy-number variations (CNVs) (Figure 2A and Appendix A). Among them, the BRCA1 (chr17q21.31) and BRCA2 (chr13q13.1) loci were frequently amplified (CNV-gain), and most were shared between primary and metastatic tumors. Since BRCA1/BRCA2 are frequently mutated in PanNETs and are significant predictors of survival outcome in pancreatic cancer patients [22], we analyzed if BRCA1/2 CNVs could impact the PFS in PanNETs patients. Although the Kaplan–Meier analysis revealed no statistically significant difference in PFS between patients with and without BRCA1/2 CNVs, these mutations seem to confer a better prognostic outcome. (37.0 months, 95% CI, 22.5-52.7 vs. 24.0 months, 95% CI, 20.6-28.7; hazards ratio, 1.542, 95% CI, 0.471-5.052; *p* = 0.074) (Figure 2B). We reasoned that the lack of statistical significance could be due to the relatively small sample size, and we believe that the observation warrants further studies. 

Structural variant (SV) analysis revealed that MITF, the microphthalmia-associated transcription factor, was observed both in PanNET and liver metastasis, with a rate of 36.4% (Appendix A). DAXX deletion was also seen in one patient and a high impact frameshift variant (c.207+1G>A) was identified in another patient (Figure 2A and Appendix A). Finally, no evidence of microsatellite instability and germline or somatic mutations in MSI-related genes were identified in the 11 patients.

### 3.3. Spreading Routes of PanNETs in One Special Patient

To characterize the spatiotemporal metastases patterns of PanNET, we first reconstructed the clonal evolutionary history and metastatic routes for one special patient (Case No. 11), who was diagnosed with liver, ovarian, and brain metastases. Between February 2014 and July 2019, the patient received pancreatectomy for removal of primary PanNET, hepatectomy for liver metastasis and bilateral ovariectomy for ovarian metastasis at an interval of two to three years between surgeries (Figure 3A). Of note, the primary tumor was G2, liver metastasis was NET G3, and the ovarian metastasis was G1 in this patient (Appendix A). To understand heterogeneity and the evolutionary trajectories of PanNET progression in this patient, we re-sequenced the patient’s primary tumor (*n* = 1), liver metastases (*n* = 2), ovarian metastases (*n* = 2), and blood sample via WES. 

WES revealed high genomic heterogeneity between the primary tumor and its liver and ovarian metastases, including CNVs and SVs, between metastases of different organs, as well as between different metastatic sites within the same organ (Figure 3B–D and Appendix A), which could be due to the long interval between the primary and metastatic tumors; in the process of tumor growth after metastasis, novel mutations are constantly acquired, which increases the genetic heterogeneity between them. Apart from intratumor heterogeneity, these results revealed spatiotemporal heterogeneity among the primary tumor and its metastases which formed monophyletic clades, suggesting that liver metastasis and ovarian metastasis were founded by different populations of cells (Figure 3E). In combination with the clinical course of case No. 11, we hypothesize that, at least in this patient, distant dissemination seeding occurred before the first operation.

### 3.4. Clonal Cluster of PanNETs 

To further explore the clonal evolutionary from primary PanNETs to liver metastasis, we reconstructed the clonal evolutionary history and metastatic routes for other patients. Pyclone was adopted to calculate the cancer cell fractions (CCFs) of each mutation, which were then grouped into mutation clusters (Figure 4).

Here, we first demonstrate our analysis strategy with Patient 01, who harbored 222 mutations in primary tumor and 194 mutations in metastasis tissue, with 173 shared mutations. Eight mutation clusters were identified in this patient (Figure 4). Different clusters displayed distinct CCFs in both primary tumor and liver metastasis, which is indicative of constant, polyclonal seeding. As for Patient 01, the blue cluster increased significantly from primary tumor to liver metastasis, suggesting that the mutations in this cluster (e.g., GNAQ and MED12) had a selective advantage that was possibly associated with niche outgrowth in the distant site. Similar trend was observed in all other patients. These results suggest that polyclonal seeding was the major form of metastasis in these PanNET patients.

## 4. Discussion

Clonal evolution is an important biological process in metastatic progression of PanNETs, but the magnitude and the clinical relevance of genomic heterogeneity PanNETs metastasis remains largely unexplored. In the current study, we employed NGS to delineate the mutational characteristics and intratumoral heterogeneity of primary PanNETs and paired metastases. This study provides the first direct piece of evidence for the presence of a high degree of shared mutations between PanNETs and liver metastases, and reveals that liver metastases in PanNETs patients have a polyclonal origin. Consistent with other findings in breast and lung cancer metastasis [23], polyclonal seeding in PanNETs metastases was associated with a less favorable survival outcome. Polyclonal seeding likely provides the metastases with the necessary genetic and phenotypic diversity that confers competitive advantages with respect to growth, expansion as well as resistance to treatment. For instance, shared clonal mutations in driver genes such as ATM, ATRX, BRCA1/2, DAXX, MSH3, and MSH6 are associated with aberrations in major cellular signaling pathways like Pathways in cancer, DNA damage response, and regulation of cell cycle. Changes in these important cellular signaling pathways could be of fundamental significance in driving tumor progression to metastasis, and emergence of drug resistance, consequently resulting in poor survival outcomes. Furthermore, our findings indicate that clonal mutations in metastases constitute distinct subsets of gene mutations that occur early during oncogenesis of PanNETs, suggesting that the metastatic potential of PanNETs was specified early. This warrants a need to target the driver mutations of tumorigenesis in PanNETs. The results also indicate that metastatic cells continually accrued new private mutations that further differentiate themselves from primary tumor cells genetically, and these private mutations might be required for niche independence and growth. In view of the fact that primary tumor was resected prior to detection of liver metastases as well as ovarian metastases, and a high percentage of mutations were shared among the primary PanNETs and liver and ovarian metastases, it remained a distinct possibility that liver metastasis occurred as early as or prior to the resection of the primary tumor as liver metastases may be too small to be detected at the time of surgery. 

It was reported that DAXX and MEN1, which are involved in chromatic remodeling, were mutated in PanNETs [24]. This study also found that DAXX and MEN1 were mutated both in the primary tumor and liver metastasis. In addition, we found that other genes including SETD2, ARID2, and KMT2C that are involved in chromatin organization were mutated in PanNETs and liver metastases. This suggests that aberrant chromatin remodeling is an important pathongenic mechanism in both primary PanNETs and liver metastases. Previously, Roy et al. found loss or deletion of DAXX and disruption of SETD2 function in 81% of primary PanNETs with distant metastases and showed that these genomic changes were associated with shorter disease-specific survival [25]. Similarly, Cives et al. further showed that mutations in DAXX in PanNETs were associated with increased grade, lymphovascular invasion, and reduced disease-free survival [26]. Taken together, these findings indicate that alterations in chromatin-remodeling machinery may contribute to metastasis of PanNETs and may be used as biomarkers in predicting malignant progression of PanNETs. Furthermore, in this study, we discovered recurrent shared clonal somatic mutations in BRCA1 and BRCA2. In light of this, we posit that PARP inhibitor, olaparib, which is conventionally used as maintenance treatment for cancer patients who have deleterious germline BRCA1/2 mutations, may be used to treat metastatic PanNET patients with somatic BRCA1/2 mutations. This study also uncovered several missense variants of ARID2 in both primary PanNETs and liver metastases. Though mutations in ARID2 have been described in pancreatic cancer [27], no mutations of ARID2 have been previously reported in primary PanNETs and liver metastases. In addition, this study also provides the first report of MITF deletion in PanNETs and liver metastases. MITF, the microphthalmia-associated transcription factor, is involved in multiple cellular processes including cell survival and proliferation, invasion, and DNA damage repair [21]. It also activates the expression of INK4A by binding to its promoter and induces retinoblastocyte protein (Rb) to be underphosphorylated, leading to cell cycle arrest and differentiation into terminal cells [28]. All in all, this study provides a list of metastasis related somatic mutations that may aid in the development of treatment strategies.

Notwithstanding, this study has several limitations. The sample size was rather limited; only 11 patients with PanNETs with liver metastasis were analyzed in the study. In addition, though the patients were collected from a prospectively maintained database at our institution, the study was retrospective in nature and was carried out at a single tertiary care institution. Moreover, no functional verification experiment was performed. Nonetheless, this study provides valuable insights into the molecular mechanisms underlying metastatic progression in PanNETs.

## 5. Conclusions

In conclusion, we showed that liver metastases were polyclonal in nature, as evidenced by the high degree of shared clonal mutations. These polyclonal metastases are associated with worse prognostic outcome. Furthermore, we found evidence that suggests that gene mutations that potentiate metastases might have been acquired early during tumorigenesis of PanNETs. Our study also uncovers some novel gene mutations that could be further explored for therapeutic manipulation and prognostic significance for PanNETs.

## Figures and Tables

**Figure 1 genes-13-01588-f001:**
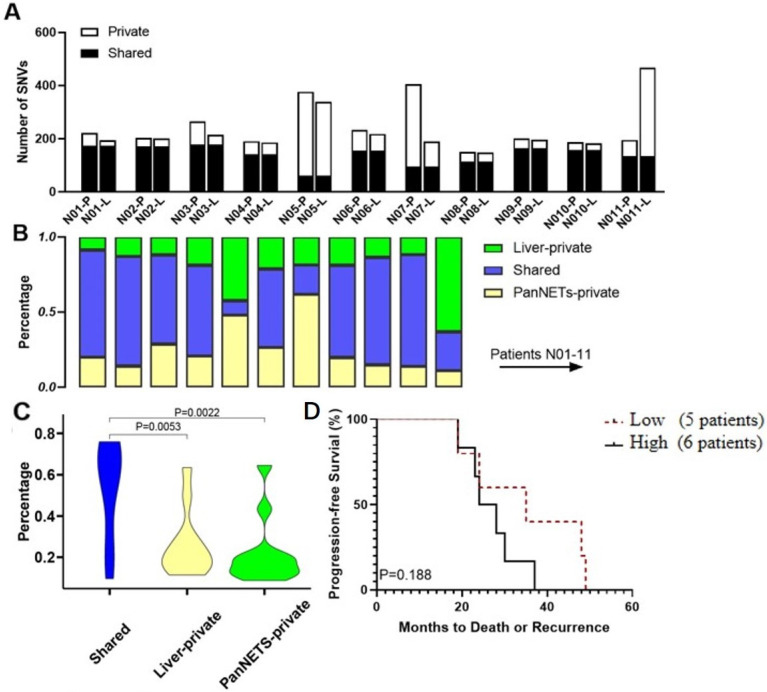
Genomic heterogeneity in paired primary pancreatic neuroendocrine tumors (PanNETs) and liver metastases. (**A**) The number of single nucleotide variants (SNVs) in each tumor tissue is shown. (**B**) The percentage of identified SNVs that are shared or private in PanNETs and liver metastases. (**C**) Violin plots illustrate the statistical results of the relative percentage of shared, primary PanNETS-private and metastasis-private SNV. The *p* value was calculated by Student’s test (*t*-test). (**D**) The Kaplan–Meier curve of progression-free survival (PFS) of the study patients stratified by a low (*n* = 5) vs. high (*n* = 6) proportion of common SNVs and indels. The *p* value was calculated by the log-rank test.

**Figure 2 genes-13-01588-f002:**
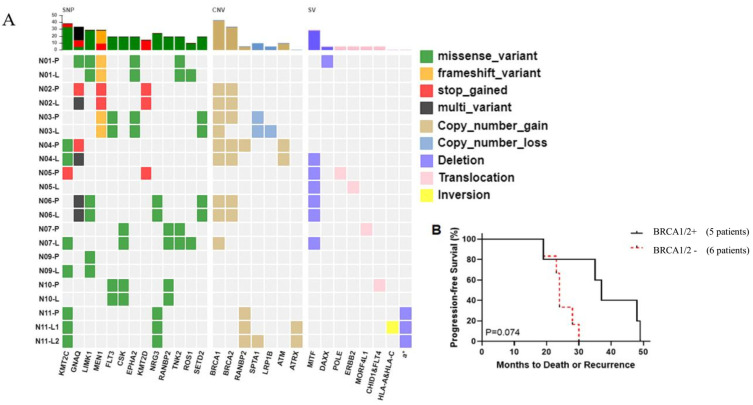
The mutational landscape in paired primary PanNETs and liver metastases. (**A**) Potential driver alterations identified in paired primary PanNETs and liver metastases, a*: HLA−B & HLA−C & RPL3P2 & USP8P1 & WASF5P & XXbac-BPG248L24.10 & XXbac-BPG248L24.12 & XXbac-BPG248L24.13. (**B**) The Kaplan–Meier curve of PFS of the study patients stratified by positive (*n* = 5) or negative (*n* = 6) BRCA1/2 CNV. The *p* value was calculated by the log-rank test.

**Figure 3 genes-13-01588-f003:**
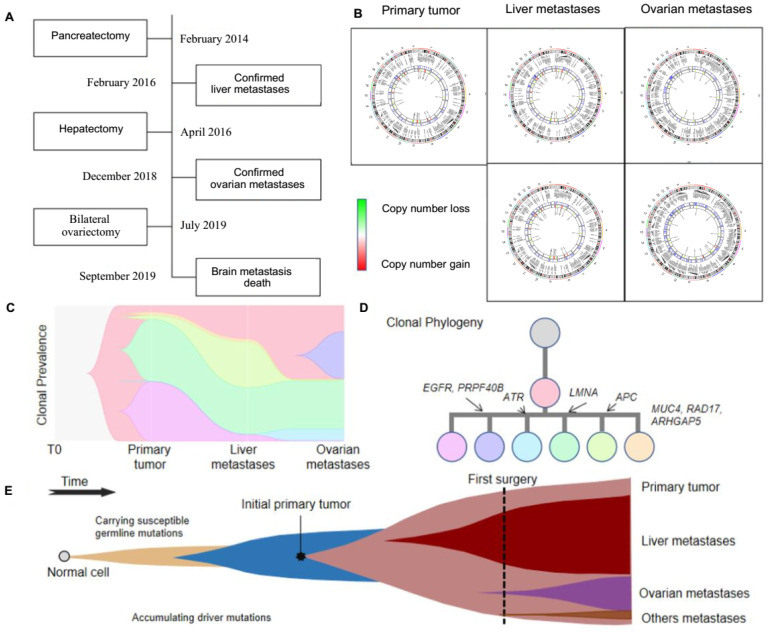
Evolutionary trajectories of PanNETs in a patient with liver and ovarian metastases. (**A**) Case No.11 underwent three operations within five years to remove the primary tumor, liver metastases, and ovarian metastases, respectively. Her five tumor samples and blood were whole-exome sequenced. (**B**) Circos plots display the chromosomal distribution of variations in the five tumor samples from the patient. The outer rings indicate SNVs and the inner rings indicate copy number variations (CNVs). The high-resolution version of the circos plots can be found in Appendix A. (**C**) The dynamic diagram of clonal prevalence during the dissemination from primary PanNET to the liver and ovarian. (**D**) Tumor phylogenies are reconstructed based on somatic variations of the patient. (**E**) Schematic illustration of tumor evolution: starting with normal cells that carry susceptible germline mutations, more and more driving mutations are accumulated as the cells proliferate, leading to malignant transformation, growth of the primary tumor and metastatic dissemination, seeding and outgrowth.

**Figure 4 genes-13-01588-f004:**
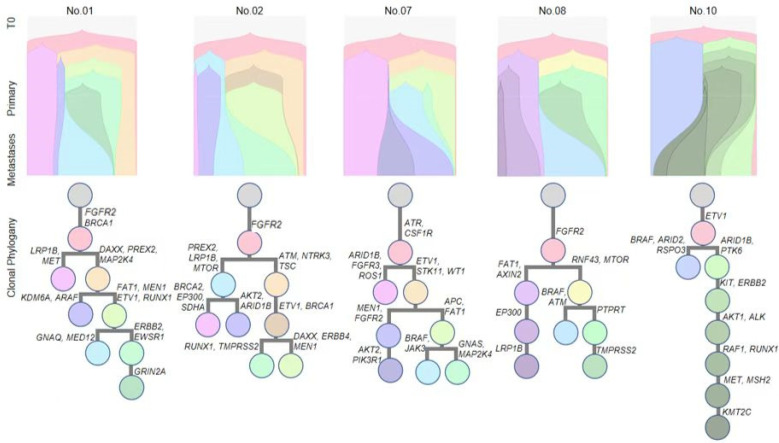
Clonal evolutionary structures inferred from the subclonal structure of 5 characteristic cases. We reconstructed the clonal evolutionary history and metastatic routes for other patients. Pyclone was adopted to calculate the cancer cell fractions (CCFs) of each mutation, which were then grouped into mutation clusters.

**Table 1 genes-13-01588-t001:** Demographic and baseline characteristics of patients with pancreatic neuroendocrine tumors and liver metastases.

No.	Sex	Age Years	Primary PanNET				Liver Metastases		Positive Lymph Nodes	Nerve Invasion	Lymphovascular Invasion	Outcome
			Size (cm)	Location	Grade	Ki67	Size (cm)	Grade				Months
**1**	M	42	5.1 × 4.0 × 3.0	Head	G2	15%	3.0 × 1.0	G2	4/8	+	−	PFS: 30 M
**2**	M	53	8.0 × 7.0 × 4.5	Tail	G2	5%	2.0 × 1.8 × 1.8	G2	0/3	−	−	OS: 19 M
**3**	M	60	9.0 × 4.5 × 2.8	Tail	G2	4%	4.0 × 2.5 × 3.0	G2	3/8	−	−	PFS: 35 M
**4**	M	67	3.0 × 1.5 × 1.0	Head	G1	2%	0.6 × 1.0 × 1.0	G1	6/18	−	+	PFS: 37 M
**5**	F	33	6.0 × 5.0 × 4.5	Tail	G2	10%	3.5 × 2.5 × 1.5	G2	1/11	−	+	PFS: 19 M
**6**	F	44	4.0 × 2.7 × 2.5	Tail	G2	5%	1.5 × 1.2 × 1.0	G2	0/10	−	−	PFS: 48 M
**7**	F	47	3.0 × 2.5 × 1.8	Head	G1	<2%	2.0 × 1.5 × 2.0	G1	3/7	+	−	PFS: 49 M
**8**	F	53	4.5 × 3.0 × 2.5	Tail	G2	5%	5.5 × 4.5 × 4.5	G2	0/4	−	+	PFS: 28 M
**9**	F	59	5.5 × 4.4 × 2.5	Tail	G2	4%	3.5 × 1.0	G2	2/2	−	+	PFS: 23 M
**10**	F	61	1.9 × 1.0 × 0.2	Body	G2	4%	Multiple metastasis (Max: 1.0 × 0.8)	G2	0/3	−	−	PFS: 24 M
11	F	46	6.0 × 5.0 × 3.5	Head	G2	10–20%	Bilateral: 4.0 × 3.0 (left) and 3.0 × 2.5 × 2.0 (right)	G3	0/29	−	−	PFS: 24 M

G1: low grade, G2: intermediate grade, and G3: high grade; “+”: positive, “−”: negative. Abbreviations: OS: overall survival, PFS: progress-free survival.

**Table 2 genes-13-01588-t002:** Mutations in genes associated with PanNET are known.

Gene No.	Mutated Genes	Mutations Nucleotide	Mutations Protein	Mutation Type	Mutations Tissue (Patient No.)	
					Primary	Metastases
1	APC	c.6973G>A	p.Gly2325Ser	missense_variant	-	5
		c.2098G>T	p.Asp700Tyr	missense_variant	5	-
		c.1412G>A	p.Gly471Glu	missense_variant	7	-
		c.3341G>A	p.Arg1114Gln	missense_variant	7	-
		c.6857C>T	p.Ala2286Val	missense_variant	7	-
		c.3949G>C	p.Glu1317Gln	missense_variant	9	9
2	ARID2	c.4300G>T	p.Ala1434Ser	missense_variant	10	10
		c.1759A>G	p.Ser587Gly	missense_variant	4	4
		c.929G>A	p.Arg310His	missense_variant	5	-
		c.1368G>A	p.Met456Ile	missense_variant	5	-
		c.4300G>T	p.Ala1434Ser	missense_variant	10	10
3	ATM	c.821C>A	p.Ser274Tyr	missense_variant	-	2
		c.8120C>G	p.Ser2707Cys	missense_variant	2	2
		c.6115G>A	p.Glu2039Lys	missense_variant	-	6
		c.497-4delT	-	frameshift_variant	3	-
		c.2466+7A>G	-	frameshift_variant	6	6
		c.1339C>T	p.Arg447*	stop_gained	7	-
4	BRCA1	c.5636T>C	p.Ile1879Thr	missense_variant	7	-
		c.3448C>T	p.Pro1150Ser	missense_variant	4	4
		c.1775G>A	p.Ser592Asn	missense_variant	-	5
		c.3167C>G	p.Ser1056Cys	missense_variant	5	-
		c.4046C>T	p.Thr1349Met	missense_variant	6	-
		c.2875A>G	p.Arg959Gly	missense_variant	-	8
		c.5314C>T	p.Arg1772*	stop_gained	7	-
5	BRCA2	c.8187G>T	p.Lys2729Asn	missense_variant	1	1
		c.4585G>A	p.Gly1529Arg	missense_variant	2	-
		c.10234A>G	p.Ile3412Val	missense_variant	3	3
		c.1012G>A	p.Ala338Thr	missense_variant	6	-
		c.9836T>C	p.Leu3279Ser	missense_variant	7	-
		c.9139C>T	p.Gln3047*	stop_gained	-	5
6	DAXX	c.1111C>T	p.Arg371Trp	missense_variant	2	2
		c.207+1G>A	-	frameshift	9	9
7	MSH3	c.181_189dupGCAGCGCCC	p.Ala61_Pro63dup	conservative_inframe_insertion	10/4/6	10/4/6
		c.2071G>A	p.Glu691Lys	missense_variant	-	7
		c.356C>T	p.Ser119Phe	missense_variant&splice_region_variant	1	1
		c.1764-1G>A	-	frameshift_variant	-	7
8	MSH6	c.4068_4071dupGATT	p.Lys1358fs	frameshift_variant&stop_gained	4	4
		c.3557-4delT	-	frameshift_variant	5/7	5
9	PALB2	c.925A>G	p.Ile309Val	missense_variant	2	2
		c.1571C>T	p.Ser524Leu	missense_variant	-	5
10	RAD50	c.3697C>A	p.Pro1233Thr	missense_variant	7	-
11	RAD51	c.88C>T	p.Gln30*	stop_gained&splice_region_variant	5	-
12	RB1	c.2393G>A	p.Arg798Gln	missense_variant	-	5
		c.1597G>A	p.Glu533Lys	missense_variant	5	-
		c.2729G>A	p.Arg910Gln	missense_variant	-	7
		c.1422-9_1422-8delTT	-	frameshift_variant	3/4/5	3/5
13	SETD2	c.3382delA	p.Thr1128fs	frameshift_variant	3	-
		c.578C>T	p.Pro193Leu	missense_variant	3/6	3/6
		c.4162G>T	p.Asp1388Tyr	missense_variant	7	-
14	SMARCA4	c.113C>G	p.Ser38Cys	missense_variant	-	5
		c.2381C>T	p.Thr794Met	missense_variant	5	-
		c.2620C>T	p.Arg874Cys	missense_variant	6	-
15	TSC1	c.3124_3129delAGCAGC	p.Ser1042_Ser1043del	conservative_inframe_deletion	-	9
		c.3114C>A	p.Ser1038Arg	missense_variant	7	-
		c.1438+6G>A	-	frameshift_variant	5	-
16	TSC2	c.3385C>T	p.Arg1129Cys	missense_variant	2	2
		c.856A>G	p.Met286Val	missense_variant	3	3
		c.202G>A	p.Ala68Thr	missense_variant	4	-
		c.5251C>T	p.Arg1751Cys	missense_variant	-	6
		c.2962C>T	p.Arg988Cys	missense_variant	7	-
		c.3412C>T	p.Arg1138*	stop_gained	9	9

**Table 3 genes-13-01588-t003:** The analysis of KEGG and GO.

Category	Primary	Metastases
KEGG Pathway	Pathways in cancer	Pathways in cancer
	Melanoma	EGFR tyrosine kinase inhibitor resistance
		Transcriptional misregulation in cancer
GO Biological Processes	peptidyl-tyrosine phosphorylation	positive regulation of transferase activity
	phosphatidylinositol-mediated signaling	transmembrane receptor protein tyrosine kinase signaling pathway
	regulation of cellular response to stress	regulation of DNA metabolic process
	regulation of DNA metabolic process	negative regulation of cell proliferation
	negative regulation of cell proliferation	apoptotic signaling pathway
	DNA repair	negative regulation of cell cycle
		epithelial cell proliferation

## Data Availability

The data presented in this study are available on request from the corresponding author.

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
