# Peer review of "Evolutionary Trajectories of Primary and Metastatic Pancreatic Neuroendocrine Tumors Based on Genomic Variations"

_genes, 2022, doi:10.3390/genes13091588_

Round 1
Reviewer 1 Report
The authors investigated the mutational landscape of eleven primary-metastatic pancreatic neuroendocrine neoplasms (PanNET) tissue pairs by an NGS panel and discovered the clonal evolution of cancer cells. The idea is interesting, the methods are suitable, the results sound acceptable, and the manuscript is well-written. In addition, I really enjoyed the illuminating and attractive figures, in particular those regarding the spatiotemporal assessment of patient 11.
Below please find my further comments:
Comments:
· I could not find the supplementary files that are referred to in the text.
· Abstract: I suggest omitting the words “KEGG” and “GO” which are in parentheses.
· Introduction: lines 38-40: according to the 2017 WHO blue book of endocrine tumors and AJCC 8th edition, not all grade 3 NETs are classified as NECs. In fact, well-differentiated NETs can also be classified as grade 3 if the mitosis counts per 10 HPF/ki67 index > 20(%), and the term NEC/poorly-differentiated should be conserved for high-grade tumors that have atypical cells (whether small or large).
· Line 85: Please include the full form of FUSCC because here is the first time it appears in the text.
· Line 98: Metastatic tissue is not mentioned for the frozen specimens.
· Line 106: Please include the NGS panel's name so that everyone can find its specifications such as the list of the included genes and its strengths and probable shortcomings. I especially would like to know whether it is validated for microsatellite instability detection (line 244).
· Please mention the software(s) for the analyses (I guess all were performed in R).
· Line 125: Please include the full form of IPA.
· Was the functional type of PanNETs known, such as the presence of carcinoid syndrome? It is good information to add if available.
· Table 1. What does “Bilateral” mean in the liver metastases part for case 11?
· Fig. 1C: Please include the name of the statistical test used.
· Fig. 1D and its associated text: I wonder whether the authors compared the private genes and their variants between low and high groups. The genes/variants involved may confound the findings.
· Lines 196-201, Fig. 2: Twelve genes are mentioned but the number of involved genes is written to be 11. I think this discrepancy is related to the NRG3 gene which is written in line 200 but is absent in Fig. 2. What is more, I wish to know the details for these variants. I guess they are in the supplementary Table 1.
· Table 2: Although the c.578C>T variant in the SETD2 gene was present in two primary-metastasis pairs (patients 3 and 6), it was not considered a recurrent SNV.
· Fig. 2: Please review Fig. 2A. KMT2C variants are only shared in one primary-metastasis pair but it was regarded as a recurrent gene; The top colored bars for SNPs do not fully correspond to the underneath colors, for instance, the bar for CSK gene is mostly red but no stop-gain was recorded, or the reverse is true for the KMT2D gene where stop gains are illustrated but the top bar shows a high frequency of frame-shifts.
· Line 242: The c.207+1G>A variant in the DAXX gene is mentioned as a frameshift variant but it was curated as "splice_donor_variant & intron_variant" elsewhere (in Table 2). I suggest keeping consistency.
· Line 256: blood sample was left unaddressed in the text. Did the authors use it for germline investigation?
· Fig. 3. The circus plots are too small, rendering them uninformative. I suggest presenting a larger version of them in a supplementary file as well.
· Line 288: it is mentioned as “four” but I can count eight mutation clusters for case No. 1 in Fig. 4.
· Lines 291-292: “(list some relevant mutations)” looks like a reminder note.
Reviewer 2 Report
The authors of “Evolutionary trajectories of primary and metastatic pancreatic neuroendocrine tumors based on genomic variations” analysed matched primary tumours and liver metastases from 11 patients with pancreatic neuroendocrine tumour using targeted gene sequencing to identify mutations and copy number aberrations with the goal to see genomic differences between primary and metastatic tumours as well as to observe the evolutionary trajectories of this disease.
This study is interesting and could provide some additional insights of the evolution of this this tumour type. Given also that pancreatic neuroendocrine tumours are rare, makes the results of this study valuable. The main issue of this study is the small cohort size that makes any statistical analysis problematic, but I can understand the difficulty of obtaining matched primary and metastasis tumours.
I have several questions and comments/suggestions for the authors that, as I believe, could help improve the manuscript:
1) In the Material and Methods section, please cite all the software/tools used in this study, as none was cited.
2) Also, it would be good to add software versions (like it was done for the FastQC software).
3) Additionally, please write if the default parameter setting were used or specific settings were changed in the tools used.
4) Was there a specific reason why the authors used VarDict for calling mutations instead of other tools? Was there any threshold used for filtering mutations?
5) I would suggest having some consistency across the text whether you call the metastasis “liver” or “hepatic”.
6) Lines 174-185, I would be careful with having any conclusions with such a small cohort. The p-value is not significant and even if it was, 5 vs 6 patients is small size to make robust and reproducible conclusions.
7) Figure 1C: please add the x axis labels for easier reading.
8) Figure 1D: please add the number of patients in each group you compare, either in the figure or in the figure legend, so that the reader can evaluate the results.
9) Also Figure 1 legend C: please add the word “p-value” such as “** p-value <0.01, *** p-value < 0.001”
10) Line 207, remove the second “repair” word in “DNA repair “repair”.
11) I would prefer to move table 2 to the supplementary materials (although I leave this up to the decision of the authors).
12) Figure 3A: In the last box, please write “Brain metastasis and death” and not “died” as it is now.
13) Figure 3B: the circus plots are not readable. I guess it’s the image conversion, please fix.
14) Figure 3D: Correct “clonal phylogany” title
15) Line 290: Correct “bule” Cluster with blue cluster
16) Line 291-292: Indeed “list some relevant mutations”!
17) Regarding Figure 4, it would be valuable for the reader and since the cohort is small, to make the same plots for all patients and not only the selected 5 (the other plots can be added in the supplementary material).
18) Figure 4: Correct “Clonal Phylogany”
19) Lines 326:328: Again, the results were not significant, and the cohort size is small, this needs to be mentioned.
20) Lines 335-336: Please use the gene name (I guess the authors mean MEN1, as mentioned earlier).
21) Please add a data availability statement and if possible, make the raw data available.
Round 2
Reviewer 1 Report
Thank you for the revisions.